# Synthesis, Characterization and Properties of Biodegradable Poly(Butylene Sebacate-*Co*-terephthalate)

**DOI:** 10.3390/polym12102389

**Published:** 2020-10-16

**Authors:** Sun Jong Kim, Hyo Won Kwak, Sangwoo Kwon, Hyunho Jang, Su-il Park

**Affiliations:** 1Department of Packaging, Yonsei University, Wonju, Gangwon 26493, Korea; 7.7.7.8@yonsei.ac.kr (S.J.K.); ksw0089@naver.com (S.K.); whyhyun@naver.com (H.J.); 2Department of Agriculture, Forestry and Bioresources, Seoul National University, Seoul 08826, Korea; bk0502@snu.ac.kr

**Keywords:** biopolymer, biodegradable polyester, aliphatic-aromatic random copolyester

## Abstract

In this study, poly(butylene sebacate-*co*-terephthalate) (PBSeT) was successfully synthesized using various ratios of sebacic acid (Se) and dimethyl terephthalate (DMT). The synthesized PBSeT showed a high molecular weight (*M*_w_, 88,700–154,900 g/mol) and good elastomeric properties. In particular, the PBSeT64 (6:4 sebacic acid/dimethyl terephthalate mole ratio) sample showed an elongation at break value of over 1600%. However, further increasing the DMT content decreased the elongation properties but increased the tensile strength due to the inherent strength of the aromatic unit. The melting point and crystallization temperature were difficult to observe in PBSeT64, indicating that an amorphous copolyester was formed at this mole ratio. Interestingly, wide angle X-ray diffraction (WAXD) curves was shown in the cases of PBSeT46 and PBSeT64, neither the crystal peaks of PBSe nor those of poly(butylene terephthalate) (PBT) are observed, that is, PBSeT64 showed an amorphous form with low crystallinity. The Fourier-transform infrared (FT-IR) spectrum showed C–H peaks at around 2900 cm^−1^ that reduced as the DMT ratio was increased. Nuclear magnetic resonance (NMR) showed well-resolved peaks split by coupling with the sebacate and DMT moieties. These results highlight that elastomeric PBSeT with high molecular weight could be synthesized by applying DMT monomer and showed promising mechanical properties.

## 1. Introduction

Plastics are widely applied in various industrial fields and their usage is continuously increasing due to their low price and excellent physico-chemical properties. However, concerns over the depletion of petroleum resources, which represent the main raw material for plastics, are intensifying [1,2]. Additionally, since current plastic materials do not decompose in environment or under composting conditions, problems associated with plastics, such as the need for plastic waste treatment, the generation of harmful gases and soot due from plastic incineration and the harmful effects of plastics on the hormones of various organisms in the environment are rapidly increasing [3].

Therefore, the importance of developing eco-friendly biodegradable plastic materials to solve the problems of conventional plastic materials mentioned above is rapidly increasing. Two major strategies have been used for the development of eco-friendly biodegradable polymer materials. The first involves avoiding the consumption of non-renewable resources by using a polymer monomer obtained from nature [4,5,6] and the second consists of designing polymers with a biodegradable main chain [7]. Based on these strategies, poly(lactic acid) (PLA), poly(butylene succinate) (PBS) and poly(butylene adipate-co-terephthalate) (PBAT) have been used as biodegradable polymeric materials [8,9,10,11] and some non-degradable plastics have recently been gradually replaced by biodegradable plastics [12,13].

PBS is a biodegradable and compostable polyester that is synthesized via the polycondensation of succinic acid (SA) and 1,4-butanediol (BDO). A two-step process is generally used for the polycondensation: First, excess diol is esterified with a diacid to form PBS oligomers with the elimination of water. Secondly, these PBS oligomers are trans-esterified under vacuum to form a high-molecular-weight polymer. Because of its excellent processability, which originates from its low melting point and controllable melt flow index (MFI), PBS has been applied in eco-plastics such as mulching films, compostable bags, non-woven fabrics and catering tools that can replace non-degradable polymers [14,15,16].

Recently, research into Bio-PBS, which is produced using succinic acid obtained via the conversion of natural sugar (Bio-SA) rather than petroleum-based resources, has also been promoted. Utilizing naturally derived materials as monomers for biodegradable polymers not only prevents petroleum resource depletion but is also an attractive way to increase the utilization of various byproducts generated in the biorefinery process [17,18].

Along with SA, sebacic acid (Se), which can be obtained from castor oil, is also attracting attention as a monomer for bioplastics. It has shown promising results in biodegradation studies conducted by Organic Waste System (OWS) and other research centers, demonstrating its potential as a new material for bioplastics [19,20]. These studies indicated the possibility of using Se in bio-based and biodegradable materials for the packaging area.

Se can undergo homopolymerization to give poly(sebacic acid) (PSeA), which can be used as a tissue engineering scaffold in the biomedical field and a carrier for drug delivery in the pharmaceutical industry [21,22,23]. Additionally, Se can be polymerized with monomers capable of forming an anhydride bond to produce biodegradable plastics. The biodegradable aliphatic polymer poly(butylene sebacate) (PBSe) can also be obtained through the polymerization of Se as a dicarboxylic acid with butanediol [24]. Furthermore, to control the physicochemical properties of Se-based polymers, biodegradable poly-aliphatic/aromatic materials can also be prepared by adding a monomer with an aromatic group [23,24,25]. Poly(butylene sebacate-*co*-terephthalate) (PBSeT) with about 19,000 g/mol of molecular weight has been synthesized using different compositions of Se and terephthalic acid and the most amorphous PBSeT (5:5 sebacic acid/ terephthalic acid mole ratio) sample showed highly promising biodegradable characteristics [24]. However, previous researches reported synthesized PBSeT with relatively low molecular weights and only few mechanical property tests were conducted. And dimethyl terephthalate (DMT) has to be adequately screened as an aromatic unit, that mostly purified terephthalic acid (PTA) has been used. 

In this study, poly(butylene sebacate-*co*-terephthalate) (PBSeT) was prepared using 1,4-butanediol and sebacic acid and dimethyl terephthalate as the aliphatic and aromatic monomers, respectively. The physicochemical properties of PBSeT containing various ratios of Se and DMT were confirmed through nuclear magnetic resonance (NMR), gel permeation chromatography (GPC), tensile tests and differential scanning calorimetry (DSC) and thermogravimetric analysis (TGA). Fourier-transform infrared (FT-IR) spectroscopy was also conducted for characterization of the functional groups.

## 2. Materials and methods

### 2.1. Materials

DMT was supplied by SK Chemical (Seoul, Korea). Se and 1,4-butanediol (BDO) were purchased from Daejung Chemical&Metal Co. Ltd. (Shiheung, Korea). Titanium tetrabutoxide (TBT) was purchased from Merck Co. (Darmstadt, Germany) as a catalyst for synthesizing process.

### 2.2. Synthesis of Poly(Butylene Sebacate-Co-terephthalate) (PBSeT)

The synthesis was carried out using two different melting stages with different levels of vacuum (Figure 1). It was conducted by dosing calculated amounts of materials based on 1 mole of dicarboxylic acid. The first melting stage was the esterification stage, the second was the polycondensation stage. The esterification progressed at approximately 200–210 °C. The theoretical amount of by-products was obtained through esterification against dimethyl terephthalate and 1,4-butanediol with the 0.75 (g/mole) TBT catalyst and sebacic acid was added to the second esterification reaction against oligomer (T-B). The esterification was finished after obtaining the theoretical amount of calculated by-products. After esterification, we initiated polycondensation by increasing the temperature to 240 °C and decreasing the vacuum level (1.5 torr). All experiments were conducted using different molar ratios of sebacic acid and dimethyl terephthalate; the ratio of 1,4-butanediol to dicarboxylic acid was fixed as 1.25:1 mol/mol.

### 2.3. Film Fabrication

Films were prepared using a hot press machine (Seoul, Korea). The machine was set to 150–210 °C at 20 MPa for 2 min. Different temperatures were required due to the different DMT ratios. Therefore, the temperature for each polymer was set according to its appearance. Samples specimens were fabricated according to ISO 527.

### 2.4. NMR Analysis

^1^H NMR (500 MHz) was carried out in trifluoroacetic acid-d solution with tetramethylsilane as the reference standard using a BRUKER-AVANCE III 500 (Bruker, Karlsruhe, Germany). The ^1^H NMR peaks of the polymers showed signals characteristic of linear copolyesters, that is, resonances corresponding to the terminal and central protons of the sebacate, butanediol and terephthalate moieties. The compositions of the copolyesters were determined from the relative intensities of the peaks corresponding to the sebacic acid and terephthalate moieties.

### 2.5. Molecular Weight Analysis

The molecular weights (*M*_n_ and *M*_w_) and polydispersity index (PDI) were determined from the melt-pressed starting materials using a GPC system equipped with a Waters Alliance 2690 separation module, a Waters 484 tunable absorbance detector operating at 265 nm, an on-line multiangle laser light scattering (MALLS) detector fitted with a gallium arsenide laser (power: 20 mW) operating at 690 nm (MiniDawn, Wyatt Technology Inc., Santa Barbara, CA, USA), an interferometric refractometer (Optilab DSP, Wyatt Technology Inc.,) operating at 35 °C and 690 nm and two PLgel (Polymer Laboratories Inc., Church Stretton, UK) mixed E GPC columns (pore size range: 50–103 Å; bead size: 3 μm) connected in series. THF was used as the mobile phase at a flow rate of 1 mL/min. The sample concentrations were approximately 5–10 mg/mL in freshly distilled THF, with an injection volume of 100 μL. The detector signals were simultaneously recorded and the absolute molecular weights and PDIs were calculated using the software ASTRA 4.0 (Wyatt Technologies Inc.).

### 2.6. FTIR Analysis

The FT-IR absorption spectra of the samples were recorded under ambient conditions using an IFS 88-IR spectrometer (BrukerAXS GmbH, Karlsruhe, Germany). Spectra were measured from 4000 to 400 cm^−1^ at a resolution of 2 cm^−1^ and 16 scans were averaged for each sample.

### 2.7. Mechanical Property Analysis

The tensile strength and elongation at break values of PBSe, PBSeT and PBT were measured using an Instron universal testing machine (Model 3344, Instron Engineering Corp., Canton, MA, USA) at a cross-head speed of 500 mm/min at room temperature. The samples were prepared with a dumbbell-shaped cutter from films manufactured as described in ISO 527. The length of the sample between the grips of the testing machine was 33 mm, because we chose the shortest gap owing to the extremely long elongation of the samples. More than five sample measurements were conducted for each polymer and the results were averaged to obtain the mean and standard deviation.

### 2.8. Thermal Property Analysis

A melt index flowmeter (WL1400, WithLab, Seoul, Korea) was used at 190 °C. Samples were melted for 300 s and then pushed with a bar weighing 2.16 kg for 60 s.

Differential scanning calorimetry (DSC) was performed using a DSC Q2000 (TA Instruments, New Castle, DE, USA). The DSC scans were recorded under a nitrogen atmosphere in the −50 °C –200 °C temperature range at a heating rate of 10 °C/min. The melting temperature was determined from the main peak in the DSC curves in the initial scan. The glass transition temperatures were calculated as the midpoint of the heat capacity change in the DSC scans. The thermal stabilities of the co-polyesters were studied using a TA Instruments TGA-800 in a nitrogen atmosphere at a heating rate of 10 °C/min. The sample was maintained at 90 °C to remove moisture and the temperature was then increased to 800 °C.

## 3. Results and Discussion

### 3.1. NMR Analysis

To determine the copolymer composition and microstructure of the various PBSeT samples (Table 1), NMR analysis was performed; the ^1^H NMR spectra and the integration values of the comonomers are shown in Figure 2 and Table 2, respectively. In this study, NMR spectra of poly(butylene terephthalate) (PBT), PBSe and various ratios of PBSeT were obtained to clearly observe the differences among them. The spectra of PBSe and PBT consisted of five and two cleanly resolved peaks, respectively. However, the peaks of the PBSeT spectra were more complex due to the combinations of the 1,4-butanediol, dimethyl terephthalate and sebacic acid monomers. In particular, some of the peaks corresponding to 1,4-butanediol signal were split by the strong signals of neighboring groups, such as the carbonyl and terephthalate groups [23,24,25].

In the PBT spectrum, a peak at 8.25 ppm related to protons near the terephthalate benzene ring and two distinct signals at 4.65 and 2.17 ppm from the two methylene groups of the oxybutylene units originating from butanediol were clearly observed [23,26,27,28].

The PBSe spectra showed peaks corresponding to the methylene groups of the sebacate units at 1.38, 1.72 and 2.51 ppm, while those at 1.86 or 4.29 ppm corresponded to the methylene groups of the butylene units [24,29]. The PBSe NMR spectrum of Figure 2 showed a peak of –COOCH_2_ which was combined –OCH of butanediol with –COOH of sebacate at around 4.29 ppm and the –CH_2_ proton in aliphatic units of sebacate was found at around 2.51 ppm [24].

However, the copolymer PBSeT was composed of three monomers that could combine in various ways; thus, the polymerization pattern and the chemical bonding among the monomers was more diverse. In the PBSeT spectra, the peaks of the sebacate and butylene units at 1.41 ppm and 1.87 ppm were similar to those observed for PBSe and their intensity clearly varied with the sebacic acid content. In addition, peaks at 8.23 and 1.95 ppm corresponding to the terephthalate and oxybutylene present in PBT were clearly observed in all the PBSeT spectra without significant variations in their chemical shift or splitting.

However, butanediol related protons appear multiple peaks at various ppm depending on the order in which terephthalate and sebacate are combined with the two monomers during the synthesis of PBSeT. For example, the peaks at 4.6 and 2.1 ppm corresponded to butylene groups bound to two terephthalate groups (T-B-T) and were greatly influenced by the sebacate content, which affected their triplet signal shape. In addition, the peak at 4.3 ppm corresponded to butylene bound to sebacate (Se-B-Se); the shape of this peak also changed depending on the terephthalate content. 

Three major peaks corresponding to butanediol were observed in the 1.7–2.1 ppm region of the PBSeT spectra, namely, a low ppm (1.7), medium ppm (1.8) and high ppm (2.1) peak, which corresponded to SeBSe, SeBT and TBSe, respectively (Figure 2) [24]. These separated signals were detected by adding sebacic acid to PBSeT. The peaks were triplets because the signals of the 1,4-butanediol protons were easily split by those of the neighboring dicarboxylic acid [24]. These phenomena depended on the position of butanediol in the aliphatic dicarboxylic unit. Additionally, each of the butylene proton peaks exhibited changes in its intensity and integral value depending on the contents of sebacate and terephthalate. These triplets were found by differently combined sequences of TBT (fTT), TBSe (fTSe), SeBT (fSeT) and SeBSe (fSeSe); T means terephthalate, B means 1,4-butanediol, Se means sebacate.

The intensity of the terephthalate peak at 8.2 ppm increased with increasing DMT content, as shown in Table 2. And changes in the ratio of sebacic acid affected the intensity of its –OCH peak at 2.5 ppm and –CH_2_– peak at 1.3 ppm. These Changes were followed changing of ratio of aromatic unit or sebacate unit in chemical domain and it was observed as the integration of specific peak [27]. 

### 3.2. GPC Analysis

In the polymerization process, the molecular weight and the related PDI have a great influence not only on material processing but also on the physical properties of the polymer material. To elucidate the effect of the sebacate and terephthalate comonomer ratio on the molecular weight distribution of the synthesized PBSeT, GPC analysis was conducted. The GPC results are shown in Figure 3 and the resulting weight-average molecular weight (*M*_w_), number-average molecular weight (*M*_n_) and PDI values are shown in Table 3. The *M*_w_ of PBSe and all of PBSeT are between about 175,500 and 89,700 g/mol, the *M*_n_ of all of samples are between about 53,900 and 26,700 g/mol and PDI are between 2.3 and 3.4. These results indicated that the synthesized PBSe and PBSeT had higher molecular weights than those reported in previous studies [24,29]. The *M*_w_ and *M*_n_ of the synthesized polyesters were similar to those of biodegradable polybutylene adipate terephthalate (PBAT) copolymers and higher than those reported for polybutylene succinate (PBS) [30].

In addition, the DMT monomer content influenced the molecular weight and distribution of PBSeT. As mentioned above, DMT-free PBSe had a *M*_w_ of 175,500 g/mol, *M*_n_ of 53,900 g/mol and PDI value of 3.2. The high PDI value of PBSe indicated that it was more easily processible than the other samples but also that possible defects would need to be considered during the fabrication of thin films [31,32]. For all the PBSeT samples except PBSeT64, the *M*_w_ of PBSeT tended to decrease as the content of DMT increased. PBSeT28 showed a 60% decrease in *M*_w_ and 56% decrease in *M*_n_ compared to PBSe. PBSeT64 showed the highest M_n_, resulting in the lowest polydispersity value. The addition of appropriate amount of DMT may lead to the formation of more random copolymers by suppressing strong aggregation of the sebacic units [33], that was also suggested by the 1164 cm^−1^ C–O–C or C–O peak in its FT-IR spectrum (Figure 4) [34,35] and the decreased crystalline peak in its DSC curve and crystalline peaks of XRD graph. However, when the DMT content was increased from 60% to 80%, the *M*_w_ decreased drastically from 121,600 to 88,700 g/mol and the resulting PBSeT28 had the lowest *M*_n_ (26,700 g/mol) among all the samples except PBT.

Melt flow index (MFI) measurement is a common technique for describing polymer flow behavior and polymer processability. Table 3 shows the melt flow index values for PBSe and PBSeT. The MFI values of PBSeT82 and PBSeT46 were higher than that of PBSe due to their lower molecular weights. Additionally, when the DMT content was increased to 80%, the MFI value decreased sharply. This was attributed to the incomplete melting of PBSeT82 at 190 °C, which was further supported by the inability to measure the MFI of PBT, as it did not melt at 190 °C.

### 3.3. FTIR Spectra Analysis

FTIR peaks at approximately 3000 cm^−1^ were observed in all the samples and were attributed to C–H bond related to the aliphatic functional sites of the polymer chain [36,37]. The intensity of the C–H peak was the greatest for PBSe, while that of PBT was the weakest. Specifically, strong peaks were detected at 2919 and 2851 cm^−1^ in the spectrum of PBSe; the intensity of these peaks weakened with increasing DMT content [38,39]. PBT exhibited three low-intensity peaks corresponding to C–H stretching (2962, 2915 and 2848 cm^−1^). In the PBSeT samples, the C-H bond signals of the PBSe domain weakened and shifted as the DMT content increased [40,41,42]. When the DMT content was 80%, the PBSe peaks were shifted and mixed with the PBT peaks at 2958, 2927 and a new peak at 2853 cm^−1^ upon addition of DMT. Around 1700 cm^-1^ region is a carbonyl functional unit [43] and all synthesized samples showed a strong peak over this area. This carbonyl peak at 1729 cm^−1^ gradually decreased as the DMT content increased. The C–O stretching peaks at 1260–1268 cm^−1^ in PBSe and PBSeT tended to increase gradually as the content of DMT increased. The 1164 cm^−1^ peak related to the C–O stretching or C–O–C of PBSe decreased with increasing DMT content and shifted from 1164 to 1169 cm^−1^ [34,35].

### 3.4. Mechanical Properties

The tensile strength and elongation behavior of the synthesized PBSeT are shown in Figure 5. The tensile strength of PBSe was 15.3 MPa, which was significantly higher than the value of 3 MPa reported in previous research due to the lower molecular weight in previous reports [29]. The addition of 20% and 40% DMT resulted in decreased tensile strength values of 11.8 and 15.4 MPa, respectively. PBT showed the highest tensile strength score of 41.9 MPa.

Increasing the DMT ratio in the PBSe matrix affects the elongation properties of the resulting polymers. The sample with 40% DMT content showed a significantly higher elongation at break (1632%) value than the other samples but the elongation at break decreased as the DMT content was further increased beyond 40%. PBSeT46 showed approximately 1009% elongation at break; this value was similar to that of PBSeT28. However, the tensile strength of PBSeT46 was 16.4 MPa, which was higher than that of the 20% DMT sample. Adding 80% DMT dramatically decreased the elongation at break to 34.2%; this was pronouncedly lower than that of the other samples. PBT showed high stiffness, with 7.8% elongation at break. Based the elongation study results, 40% was found to be the most suitable DMT content for the PBSeT copolyesters. The elongation properties of the synthesized PBSeT64 polyester indicated that the DMT monomer was well positioned in the matrix to increase the amorphous phase, as also demonstrated from the XRD results in Figure 8. As described above, PBSeT64 showed the highest *M*_n_ and the lowest polymer dispersity index at 2.3, demonstrating its good dispersion during polymerization. Similar results have been reported for poly(butylene adipate-co-terephthalate); one study of PBAT containing various terephthalate ratios reported that a 40% ratio of terephthalate in the matrix resulted in the highest value of elongation and that adding more than 40% of terephthalate contributed to lower elongation at break [44]. 

Based on these results, the addition of a terephthalate such as DMT seems to affect the rigidity characteristics of the polymer matrix. In the stress–strain curve, PBT showed extremely low strain compared to other samples, while PBSeT64 showed the highest strain value and a relatively low stress value, indicating that PBSeT64 has good elastomeric property. PBSeT46 showed a high level of stress with high strain. These are the most appropriate characteristics for packaging film applications similar to those of poly(butylene adipate-co-terephthalate) [28,45]. 

### 3.5. Thermal Properties

Overall, the PBSe, PBSeT and PBT polymers synthesized in this study showed three or four major decomposition points in TGA/DTGA analysis (Figure 6 and Table 4). The DTGA graphs showed three or four inflection points during the heat decomposition process because of the various combinations in the random sequence of the comonomers sebacate, 1,4-butanediol and terephthalate [46]. 

The TGA graphs show that the onset temperature of thermal degradation gradually decreased with increasing DMT content (Figure 6). The addition of various ratios of DMT contributed to changes in the microchemical structure of PBSe, as shown in the NMR data in Figure 2. Consequently, the basic thermal characteristics also changed. This conversion of the chemical structure also resulted in a greater percentage weight loss at the first thermal degradation peak with DMT addition (Figure 6).

For PBSe, the onset temperature of thermal degradation was 377 °C with a primary thermal degradation weight loss of 66.4% at *T*1 (401.8 °C); it showed three major decomposition peaks in DGTA analysis. In comparison, PBT showed a thermal degradation onset temperature of 371.0 °C and a primary degradation weight loss of 86.0% at *T*1 (393.7 °C). The weight loss during the first thermal degradation for the polymers containing 20%, 40%, 60% and 80% of DMT were higher than that of the 0% DMT polymer (i.e., PBSe) at 86.6%, 84.9%, 85.1% and 80.4%, respectively. All the synthesized PBSeT samples showed a lower temperature of primary degradation than PBSe; however, it decreased slightly with increasing DMT ratio. However, a new peak appeared near 560 °C in the 20%, 40% and 60% DMT samples. This was attributed to the effect of the –COO ester functional group [47]. However, in the case of 80% DMT, this new peak was not observed due to the strong effect of the high terephthalate content. PBSe exhibited a residual weight of 2.5% at 663 °C and that of the PBSeT polymers ranged from approximately 1–4% with increasing DMT content. For PBT, a residual weight of 4% was observed at 697 °C.

DSC analysis of the synthesized samples revealed large differences among the melting temperatures of PBSe, PBT and PBSeT with various dimethyl terephthalate ratios (Figure 7 and Table 5). PBSe exhibited a melting point of 65.3 °C and the melting temperature decreased as the DMT content was increased to 20% and 40%, as shown in Figure 7. In particular, when the content of DMT was increased to 40%, melting point was observed at 29.3 and 93.3 °C; however, the Δ*H*f were very low as 1.9 J/g and 6.3 J/g, respectively. Similarly, as the content of DMT increased to 40%, the Δ*H*c signals were also remarkably getting weaker compared to the Δ*H*c enthalpies of other samples. The decreased crystallinity and weak peak intensity of Tm of PBSeT64 could be explained in terms of a change from a semicrystalline to an amorphous phase state, that is, an increase in the amorphous phase in the domain.

The melting points increased sharply to 149 and 190 °C when the DMT content was increased to 60% and 80%, respectively. Since the melting point of PBT was 220 °C, this phenomenon was thought to reflect the high melting point of the aromatic part in addition to the content of DMT. In addition to the melting point, the Tc also increased significantly to 91 and 146 °C at DMT contents of 60% and 80%, respectively. PBT showed the highest *T*c of 190 °C among the polymers.

The lowest glass transition temperature (*T*g) was observed in PBSeT64. PBSe exhibited −29.8 °C of *T*g and all of PBSeT samples showed the *T*g under −20 °C, which were similar to previously reported *T*g of synthesized aliphatic–aromatic copolymers with sebacate [24,25]. The low *T*g means that the polymers are in rubber state with possible elastomeric properties at room condition. These polymers, which had low glass transition temperatures, showed very soft properties similar to those of PBAT and poly(ethylene) [48,49].

### 3.6. Crystalline Structure

X-ray diffraction is a useful analysis method that can determine not only the crystal structure of a polymer material but also its degree of crystallization. Figure 8 showed the X-ray diffractograms of PBSe, PBT and PBSeT with different comonomer ratios. In the diffractogram of PBSe, crystal peaks are found at 21.2° and 24.7°; this is similar to the results previously reported for a poly(alkylene sebacate) polymer [50]. The main peaks of PBT appeared at 15.9°, 17.2°, 20.4°, 23.2° and 24.8°, corresponding to the crystal planes of conventional PBT. For PBSeT28 and PBSeT82, the crystal peaks of the PBSe homopolymer are predominant. Figure 8 also showed degree of crystallinity of synthesized samples. PBSe showed the highest degree of crystallinity as 64.3%, while that of PBSeT 64 was only 6.7%. It revealed that 6:4 molar ratio (Se:DMT) for synthesis of random copolymer was suitable for making polymers with mostly amorphous state domain. These results strongly support the hypothesis that the addition of DMT induces the polymerization of random copolymers by inhibiting the aggregation of sebacic acid. The generation of amorphous PBSeT was more predominant for PBSeT 64. This would have contributed to its low *T*m and *T*c values and high elongation at break value, which deviated from the trends observed for the other PBSeT polymers.

## 4. Conclusions

PBSeT was successfully synthesized using DMT, sebacic acid and 1,4-butanediol. The tensile strength decreased as the DMT content increased. However, the elongation at break increased when the DMT content was increased to 20% and 40% and that of the polymer with a DMT content of 40% was the highest at 1632%. This phenomenon indicated that the most suitable DMT content in terms of elongation at break was 40% and that excess DMT content worsens the physical properties. The onset temperature of thermal decomposition and initial thermal decomposition rate of PBSeT decreased with increasing DMT content but the melting points increased sharply to 149 and 191 °C at DMT contents of 60% and 80%, respectively. This phenomenon was attributed to the high thermal decomposition temperature of PBT and it was thought that the melting temperature naturally increased as the content of DMT increased. In this study, we investigated the effect of DMT on the physical properties by synthesizing sebacic acid and 1,4-butanediol. Although varying the DMT content from 20% to 80% did not have a large effect on the tensile strength, it showed an effect on the elongation, with the highest value being obtained at 40% DMT content. It was thought that these physical properties could be complemented in future studies by incorporating polymers with hard physical properties such as polylactic acid (PLA) and polyhydroxyalkanoates (PHAs).

## Figures and Tables

**Figure 1 polymers-12-02389-f001:**
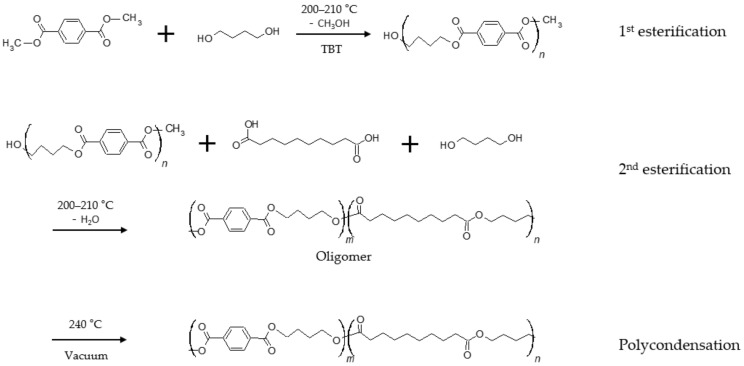
Synthetic route for the synthesis of poly(butylene sebacate-*co*-terephthalate) (PBSeT).

**Figure 2 polymers-12-02389-f002:**
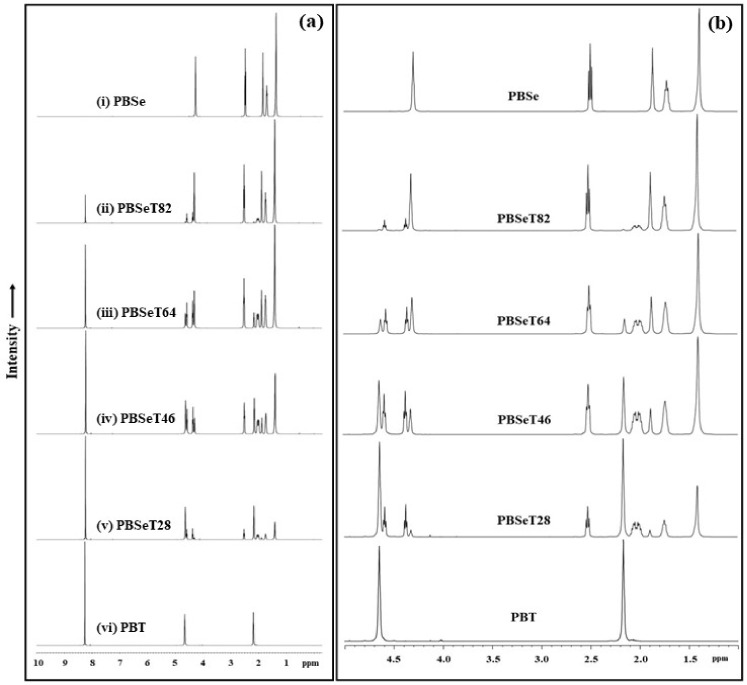
NMR data of the synthesized PBSe, PBT and the PBSeT copolymers with various sebacic acid/dimethyl terephthalate ratios. (**a**) Whole spectra and (**b**) magnification of chemical shift (1.0 ~ 5.0 ppm).

**Figure 3 polymers-12-02389-f003:**
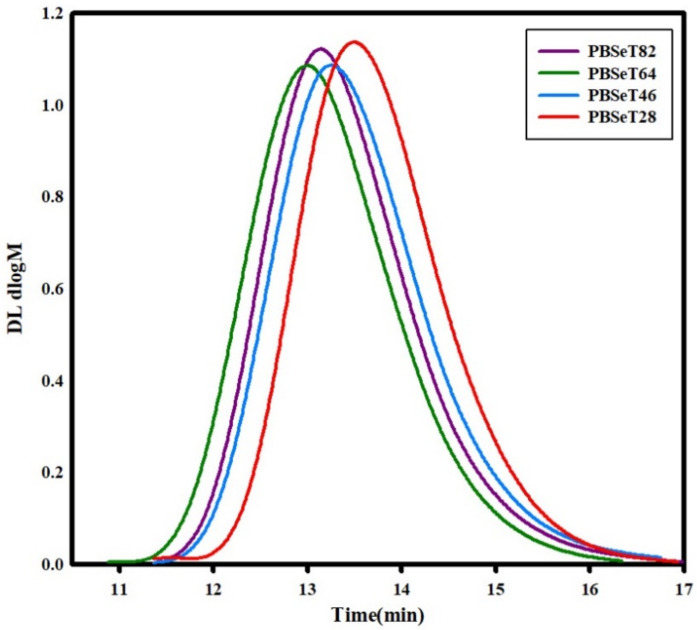
Gel permeation chromatography (GPC) plots of the synthesized PBSeT polyesters.

**Figure 4 polymers-12-02389-f004:**
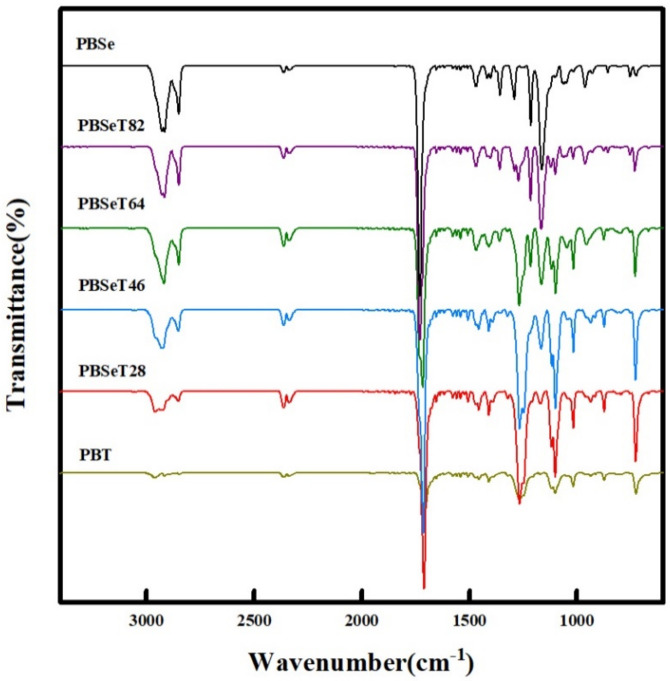
Fourier-transform infrared spectra (FT-IR) of the six different copolyester samples.

**Figure 5 polymers-12-02389-f005:**
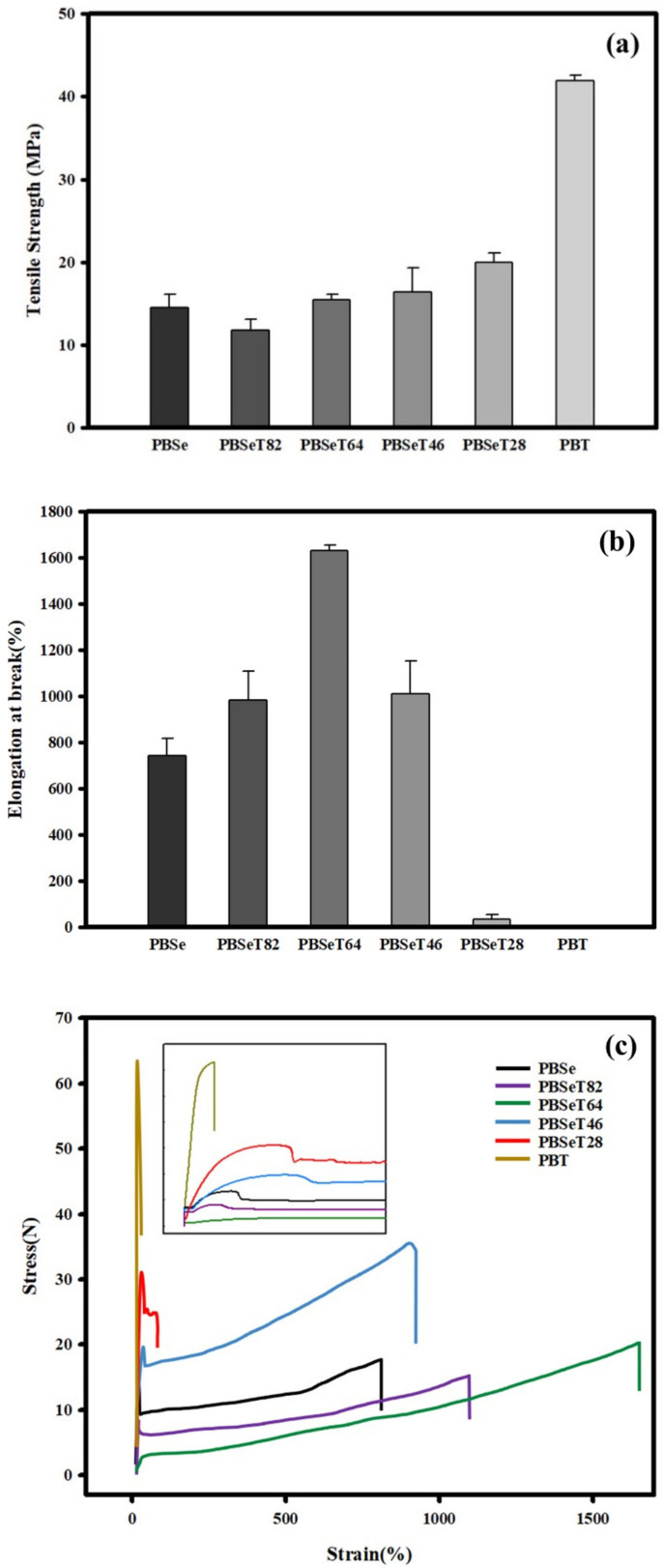
(**a**) Tensile strength, (**b**) elongation at break and (**c**) SS curve of the synthesized samples.

**Figure 6 polymers-12-02389-f006:**
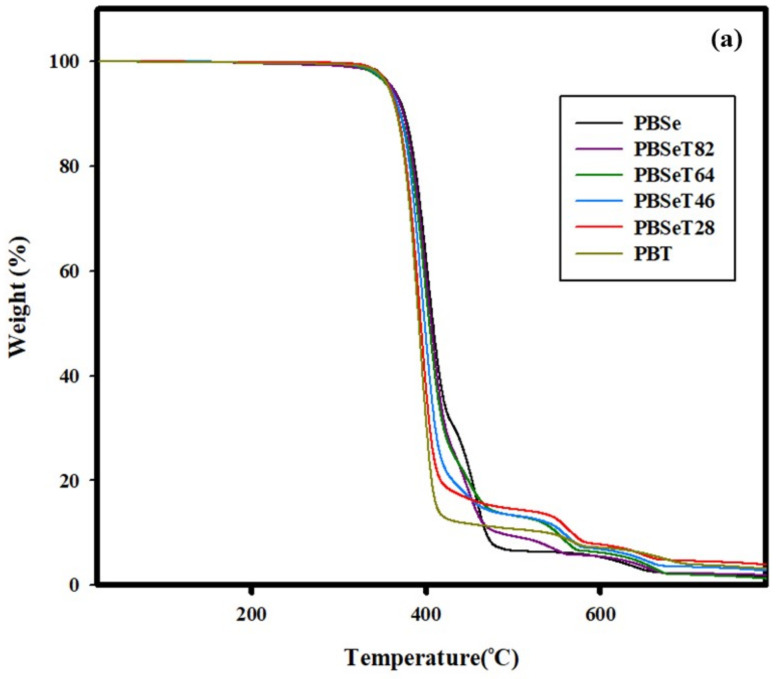
(**a**) TGA and (**b**) DTGA curves for the thermal decomposition of the copolyesters with various Se:DMT molar ratios.

**Figure 7 polymers-12-02389-f007:**
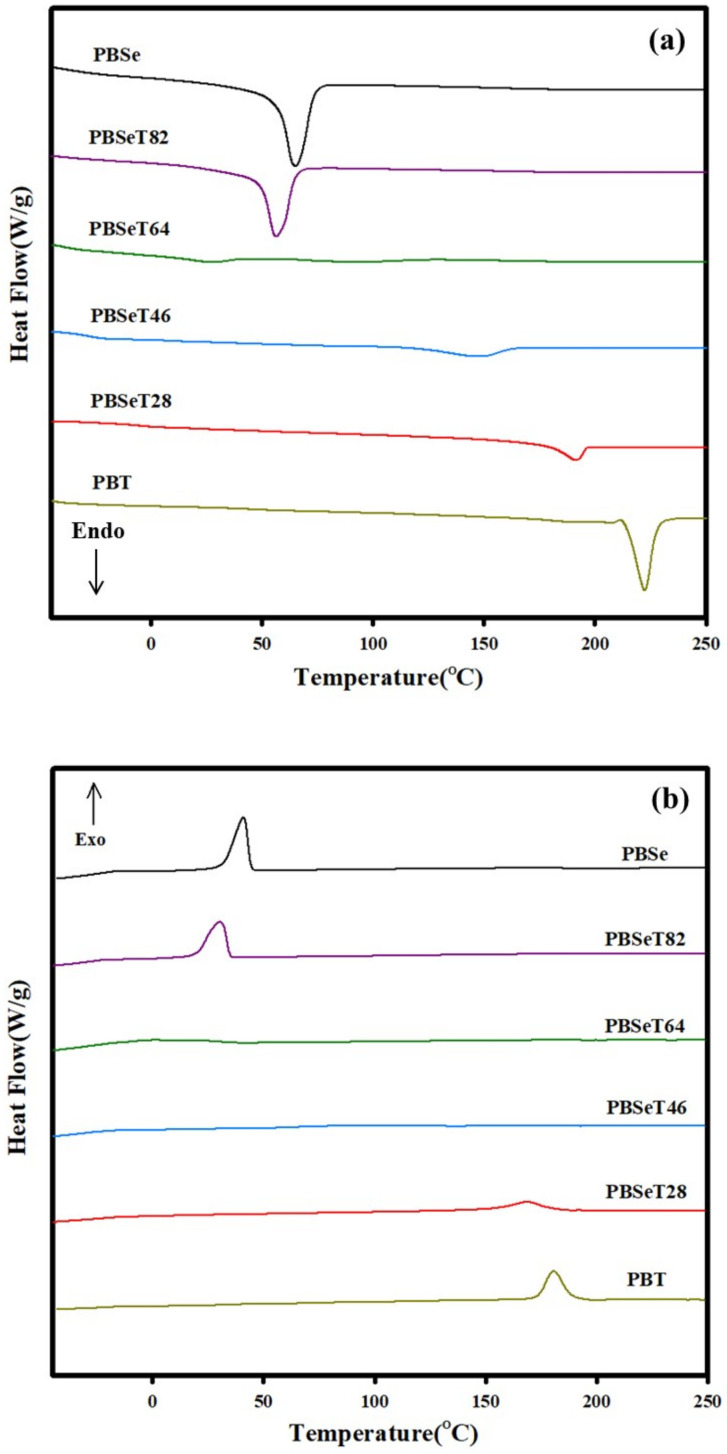
Melting point and crystallization temperatures determined from differential scanning calorimetry (DSC) for the synthesized copolyesters. (**a**) 2nd heating and (**b**) 2nd cooling.

**Figure 8 polymers-12-02389-f008:**
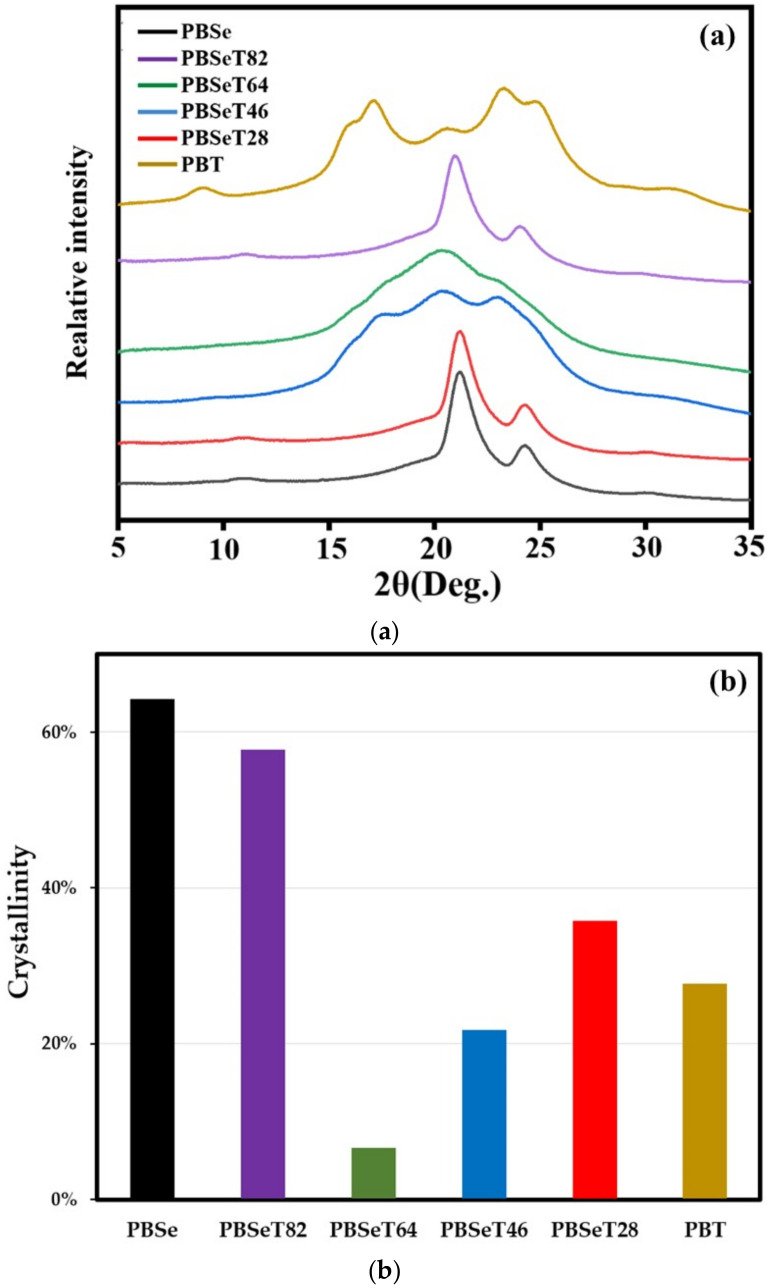
(**a**) WAXD curves and (**b**) crystallinity of the synthesized random copolymers with various sebacate:dimetylterephthalate molar ratios.

**Table 1 polymers-12-02389-t001:** Abbreviations of the studied samples with various molar ratios of aliphatic/aromatic dicarboxylic acid group.

Code Name *	Sebacic Acid (Aliphatic Dicarboxylic Acid)	Dimethyl Terephthalate (Aromatic Dicarboxylic Acid)
PBSe	100	0
PBSeT82	80	20
PBSeT64	60	40
PBSeT46	40	60
PBSeT28	20	80
PBT	0	100

* PBSeT82 means 8:2 sebacic acid/dimethyl terephthalate mole ratio.

**Table 2 polymers-12-02389-t002:** Integration value of comonomer of copolyesters with various content of aromatic units.

Sample	T at 8.2 ppm	fTT (T-B-T) at 4.6 ppm	fSeSe (Se-B-Se) at 4.3 ppm	fTT (T-B-T) at 2.1 ppm
**PBSeT82**	0.159	0.026	0.837	0.027
**PBSeT64**	0.645	0.264	0.598	0.262
**PBSeT46**	1.502	0.889	0.398	0.917
**PBSeT28**	4.018	3.075	0.216	3.219

**Table 3 polymers-12-02389-t003:** Melting flow index, polymer molecular weight and polydispersity index (*M*_w_/*M*_n_) of copolyesters.

Unit		PBSe	PBSeT82	PBSeT64	PBSeT46	PBSeT28
MFI (Melting flow index)	g (190 °C/2.16 Kg)	14	19	18.2	14	0.2
*M* _w_	g/mol	175,500	150,300	154,900	121,600	88,700
*M* _n_	g/mol	53,900	54,500	64,600	35,200	26,700
PDI(polydispersity index)	*M*_w_/*M*_n_	3.2	2.7	2.3	3.4	3.3

**Table 4 polymers-12-02389-t004:** TGA and DTGA data for the thermal decomposition of the synthesized copolyesters.

Sample	*T* Onset (°C)	*T*1 (°C)	*T*2 (°C)	*T*3 (°C)	*T*4 (°C)	Char Yield (%)
PBSe	377.6	401.8	458.9	-	642.8	2.5
PBSeT82	374.1	399.0	444.7	562.2	679.0	1.9
PBSeT64	373.8	400.1	448.3	556.5	663.3	2.1
PBSeT46	373.2	396.6	445.6	560.1	653.7	3.6
PBSeT28	370.7	393.8	-	563.7	650.8	4.9
PBT	371.0	393.7	-	567.9	677.6	4.0

**Table 5 polymers-12-02389-t005:** DSC data of the synthesized polyesters with various sebacic acid:dimethyl terephthalate ratios.

	PBSe	PBSeT82	PBSeT64	PBSeT46	PBSeT28	PBT
*Tm* (°C)	65.3	56.6	29.3 / 93.3	149.2	191.4	222.4
*ΔH**f* (J/g)	43.6	35.7	1.9 / 6.3	11.2	20.5	40.7
*Tg* (°C)	−29.8	−34.7	−38.8	−32.2	−21.3	65.3
*Tc* (°C)	41.0	30.5	22.0 / 1.5	79.0	168.6	180.7
*ΔH**c* (J/g)	68.4	58.0	3.4 / 2.8	17.4	29.6	47.7

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
