# Peer review of "Synthesis, Characterization and Properties of Biodegradable Poly(Butylene Sebacate-Co-terephthalate)"

_polymers, 2020, doi:10.3390/polym12102389_

Round 1

Reviewer 1 Report

In this paper the authors have reported the synthesis and characterization of poly(butylene sebacate-co-terephthalate). High molecular weight copolymers have been obtained and the thermal and mechanical properties of the polymers can be modified by the changing ratios of sebacate vs terephthalate. The characterization is fairly complete and the paper of well organized. Overall the paper is suitable for publication, with a few needed revisions/corrections noted below.

The understanding and interpretation of the NMR spectroscopy seems insufficient. Statements such as “some of the peaks corresponding to 1,4-butanediol were split by the strong signals of neighboring groups, such as the carbonyl and terephthalate groups”(line 153-155) are just incorrect. So is the statement “the peaks were triplets because the signals of the 1,4-butanediol protons were easily split by those of the neighboring dicarboxylic acid.”(line 188-189). In addition, the two peaks at 4.3 and 4.6 ppm in the copolymer should be described and discussed since they are related and correspond to the T-B-Se junction.

IR spectroscopy: Line 241-242: “C-H stretching (2962 cm-1) and C-H bonding (2915 cm-1 and 2848 cm-1)” here the authors make a distinction between C-H stretching and C-H bonding. Not sure what a C-H bonding is. If it meant C-H bending, then it shouldn’t show up here.  Line 250-251: “indicating that the intensity of the C-O group of sebacate and the C-OH group of DMT increased by esterification.” Not sure if I follow what this means

Line 235-287: “The DTGA graphs showed three or four inflection points during the heat decomposition process because of the various combinations in the random sequence of the comonomers sebacate, 1,4-butanediol, and terephthalate.” This seems unlikely because they appear very late during the thermal decomposition (>80%).

Line 196: here a Table 2 is referred to, but there is no “ intensity of the terephthalate peak at 8.2 ppm…” in Table 2. Line 231, Table 3 is referred to, but there is no Table 3.

Line 256: the value 20.4 MPa doesn’t match Fig 4, where it is ~15 MPa for PBSe.

Line 73-74: “a dicarboxylic acid such as  butanediol”  butanediol is not an acid

Line 95: the authors mentioned a catalyst, but what is this catalyst?

Line 94: “and” in the beginning seems to be extra.

Author Response

Reviewer 1

In this paper the authors have reported the synthesis and characterization of poly(butylene sebacate-co-terephthalate). High molecular weight copolymers have been obtained and the thermal and mechanical properties of the polymers can be modified by the changing ratios of sebacate vs terephthalate. The characterization is fairly complete and the paper of well organized. Overall the paper is suitable for publication, with a few needed revisions/corrections noted below.

Authors are pleased that reviewer found this study suitable for publication. We have modified the manuscript according to reviewer’s comment and suggestion. Thank you for your kind and valuable comments.

(R1-1) The understanding and interpretation of the NMR spectroscopy seems insufficient. Statements such as “some of the peaks corresponding to 1,4-butanediol were split by the strong signals of neighboring groups, such as the carbonyl and terephthalate groups”(line 153-155) are just incorrect. So is the statement “the peaks were triplets because the signals of the 1,4-butanediol protons were easily split by those of the neighboring dicarboxylic acid.”(line 188-189). In addition, the two peaks at 4.3 and 4.6 ppm in the copolymer should be described and discussed since they are related and correspond to the T-B-Se junction.

(Answer) There are several previously reported NMR studies on PBSeT and butanediol signal splitting [23, 25]. According to their analysis and our understanding, proton of butanediol were deshield easily by neighboring units such as dicarboxylic acid and terephthalate, then it appeared as multiplet. Several captures below are data explaining multiplet phenomena of PBSeT showing T-B-Se junction in detail. We only included T-B-T and Se-B-Se in table 2, which represent very clear increasing or decreasing trends. Additionally, two peaks at 4. 3 and 4.6 ppm representing T and Se, respectively, have been described several times (Lines 181, 184, 186, 197, 199 and also in Table 2).

Reference [23]- See attached file

Reference [25]- See attached file 

(R1-2) IR spectroscopy: Line 241-242: “C-H stretching (2962 cm-1) and C-H bonding (2915 cm-1 and 2848 cm-1)” here the authors make a distinction between C-H stretching and C-H bonding. Not sure what a C-H bonding is. If it meant C-H bending, then it shouldn’t show up here.  Line 250-251: “indicating that the intensity of the C-O group of sebacate and the C-OH group of DMT increased by esterification.” Not sure if I follow what this means

(Answer) Thank you for your corrections  We corrected the sentences taking into account your comment. For us, bonding meant bond or group.

Line 258-259: “PBT exhibited three low-intensity peaks corresponding to C-H stretching (2962 cm-1, 2915 cm-1, and 2848 cm-1)”

Line 264-265: The C-O stretching peaks at 1260–1268 cm-1 in PBSe and PBSeT tended to increase gradually as the content of DMT increased.

(R1-3) Line 235-287: “The DTGA graphs showed three or four inflection points during the heat decomposition process because of the various combinations in the random sequence of the comonomers sebacate, 1,4-butanediol, and terephthalate.” This seems unlikely because they appear very late during the thermal decomposition (>80%).

(Answer) We thought our samples showed those late decomposition peaks caused by its different random blocks. Other researcher also found those multi peaks of decomposition from their synthesized random copolyester also. Previous study captured below is for supporting our explanation.

Reference [47]- See attached file

(R1-4) Line 196: here a Table 2 is referred to, but there is no “ intensity of the terephthalate peak at 8.2 ppm…” in Table 2. Line 231, Table 3 is referred to, but there is no Table 3.

(Answer) We deeply appreciate for your detailed comment. We added Table 2 in revised manuscript.

Table 2. Integration value of comonomer of copolyesters with various content of aromatic units.

Sample

T at 8.2 ppm

fTT
(T-B-T) at 4.6 ppm

fSeSe
(Se-B-Se) at 4.3 ppm

fTT
(T-B-T) at 2.1 ppm

PBSeT82

0.159

0.026

0.837

0.027

PBSeT64

0.645

0.264

0.598

0.262

PBSeT46

1.502

0.889

0.398

0.917

PBSeT28

4.018

3.075

0.216

3.219

(R1-5) Line 256: the value 20.4 MPa doesn’t match Fig 4, where it is ~15 MPa for PBSe.

(Answer) Thank you for your detailed comment. We corrected the tensile value to 15.3 MPa.

Line 274: “The tensile strength of PBSe was 15.3 MPa, which was significantly higher than the value of 3 MPa reported in previous research due to the lower molecular weight in previous reports”

(R1-6) Line 73-74: “a dicarboxylic acid such as  butanediol”  butanediol is not an acid

(Answer) We edited the sentence according to your correction.

Lines 68-69: “The biodegradable aliphatic polymer poly(butylene sebacate) (PBSe) can also be obtained through the polymerization of Se as a dicarboxylic acid with butanediol”

(R1-7) Line 95: the authors mentioned a catalyst, but what is this catalyst?

(Answer) Titanium tetrabutoxide (TBT) was used and described in 2.1 Materials (Line 89) and included the TBT amount used (Line 95).  

Line 89: “Titanium tetrabutoxide (TBT) was purchased from Merck Co. (Darmstadt, Germany) as a catalyst for synthesizing process.”

(R1-8) Line 94: “and” in the beginning seems to be extra.

(Answer) We corrected this extra “and”. Thank you for the correction.

Line 94: “The theoretical amount of by-products was obtained through esterification against sebacic acid and 1,4-butanediol with the 0.75 (g/mole) TBT catalyst, and dimethyl terephthalate was added to the second esterification reaction against oligomer (Se-B).”

Reviewer 2 Report

The authors present a study on the synthesis and characterization of a biodegradable copolymer.

I recommend the publication of the paper. Anyhow here are some corrections and suggestions for an improvement of the text.

  1. The Mw values in the abstract (line 14) are given with unreasonable precision.
  2. The last sentence of the abstract does not indicate the significance of the work. I just states again what has been done.
  3. What is the idea of the graphs on page 1? This will not work as a TOC image.
  4. The abbreviations are not fully consistent. SA stands for succinic acid and PSA stands for poly(sebacic acid).
  5. The molar ratio of 1,4-butanediol:dicarboxylic acid is given as 1.3:1 (line 99) and 1.25:1 (line 158).
  6. The headline for table 1 (line 156) should include that it is a molar ratio. The meaning of e.g. "82" in the sample name is only introduced in the abstract. It should be also part of the main text.
  7. There is no indication what is given on the y-axis in figure 1. The sequence of the sample should be from PBSe on top to PBT at the bottom as it is done e.g. in figure 3 and other figures.
  8. The PDI values in line 206 are wrong (the correct values are given in table 2). The values given for "PBSeT" correspond to the values of PBSeT28 in table 2.
  9. In table 2 the parameters for the MFI is missing the units (190 °C, 2.16 kg). Mw and Mn are given with artificial precision. No information about errors is given.
  10. In figure 3 the y-axis is not labeled. On the x-axis it should be "wavenumber" instead of "wavelength".
  11. In table 5 the unit is missing.
  12. In figure 6 labeling of y-axes is missing again.
  13. The figure in the paper lack a consistent style regarding font style and font size. For most figures font sizes could be larger, in comparison font size in figure 7 seems gigantic. 

Author Response

Thank you for your kind and valuable comments for this study. Reviewer’s comments were very useful and helpful to understand more in detail information and discussion.

(R2-1) The Mw values in the abstract (line 14) are given with unreasonable precision.

(Answer) We corrected our manuscript. Thank you for your correction.

Line 14: “The synthesized PBSeT showed a high molecular weight (Mw, 88,700–154,900 g/mol) and good elastomeric properties.”

Line 230:

Table 3. Melting flow index, polymer molecular weight and polydispersity index (Mw/Mn) of copolyesters.

Unit

PBSe

PBSeT82

PBSeT64

PBSeT46

PBSeT28

MFI

(Melting flow index)

g

(190 °C/2.16 Kg)

14

19

18.2

14

0.2

Mw

g/mol

175,500

153,00

154,900

121,600

88,700

Mn

g/mol

53,900

54,500

64,600

35,200

26,700

PDI
(polydispersity index)

Mw/Mn

3.2

2.7

2.3

3.4

3.3

(R2-2) The last sentence of the abstract does not indicate the significance of the work. I just states again what has been done.

(Answer) The last sentence of the abstract was rewritten.

Lines 23-26: “These results highlight that elastomeric PBSeT with high molecular weight could be synthesized by applying DMT monomer and showed promising mechanical properties that were suitable for developing composites with other rigid biodegradable plastics.”

(R2-3) What is the idea of the graphs on page 1? This will not work as a TOC image.

 (Answer) We tried to present the main results of this work. Those figures (NMR) will be explained as differences of each synthesized samples for chemical structure. And XRD, DSC showed changing of state from semicrystalline to amorphous. And we edited it for better representation. If the format will not work well, we are willing to change images again.

(R2-4) The abbreviations are not fully consistent. SA stands for succinic acid and PSA stands for poly(sebacic acid).

(Answer) Thanks for your detailed comment. We changed from PSA to PSeA.

Line 65: “Se can undergo homopolymerization to give poly(sebacic acid) (PSeA), which can be used as a tissue engineering scaffold in the biomedical field and a carrier for drug delivery in the pharmaceutical industry”

(R2-5) The molar ratio of 1,4-butanediol:dicarboxylic acid is given as 1.3:1 (line 99) and 1.25:1 (line 158).

(Answer) We appreciate for your correction. We used 1.25:1.

Line 100: “All experiments were conducted using different molar ratios of sebacic acid and dimethyl terephthalate; the ratio of 1,4-butanediol to dicarboxylic acid was fixed as 1.25 : 1 mol/mol.”

(R2-6) The headline for table 1 (line 156) should include that it is a molar ratio. The meaning of e.g. "82" in the sample name is only introduced in the abstract. It should be also part of the main text.

(Answer) We added the word of “molar” to clarify the meaning.

Table 1. Abbreviations of the studied samples with various molar ratios of aliphatic/aromatic dicarboxylic acid group.

Code name*

Sebacic acid

(Aliphatic dicarboxylic acid)

Dimethyl terephthalate

(Aromatic dicarboxylic acid)

PBSe

100

0

 PBSeT82

80

20

PBSeT64

60

40

PBSeT46

40

60

PBSeT28

20

80

PBT

0

100

* PBSeT82 means 8:2 sebacic acid/dimethyl terephthalate mole ratio.

(R2-7) There is no indication what is given on the y-axis in figure 1. The sequence of the sample should be from PBSe on top to PBT at the bottom as it is done e.g. in figure 3 and other figures.

(Answer) We added y-axis on Figure 2 and the sequence of samples have been changed. Thank you for the correction

(R2-8) The PDI values in line 206 are wrong (the correct values are given in table 2). The values given for "PBSeT" correspond to the values of PBSeT28 in table 2.

(Answer) Thank you for your valuable comment. We edited it according to your comment.

Lines 222-224: “The Mw of PBSe and all of PBSeT are between about 175,500 and 89,700 g/mol, the Mn of all of samples are between about 53,900 and 26,700 g/mol and PDI are between 2.3 and 3.4.”

(R2-9) In table 2 the parameters for the MFI is missing the units (190 °C, 2.16 kg). Mw and Mn are given with artificial precision. No information about errors is given.

(Answer) We agree with reviewer’s comments. All of Mw and Mn values were corrected according to the results of analysis data. According to our finding, most papers did not include errors about Mw.

Table 3. Melting flow index, polymer molecular weight and polydispersity index (Mw/Mn) of copolyesters.

Unit

PBSe

PBSeT82

PBSeT64

PBSeT46

PBSeT28

MFI

(Melting flow index)

g(190°C /2.16Kg)

14

19

18.2

14

0.2

Mw

g/mol

175500

15300

154900

121600

88700

Mn

g/mol

53900

54500

64600

35200

26700

PDI
(polydispersity index)

Mw/Mn

3.2

2.7

2.3

3.4

3.3

(R2-10) In figure 3 the y-axis is not labeled. On the x-axis it should be "wavenumber" instead of "wavelength".

(Answer) Thank you for your comment. We modified x-axis to Wavenember(cm-1). In Line 269.

(R2-11) In table 5 the unit is missing.

(Answer) Thank you for your helpful comment. We added the unit.

Table 5. DSC data of the synthesized polyesters with various sebacic acid : dimethyl terephthalate ratios.

PBSe

PBSeT82

PBSeT64

PBSeT46

PBSeT28

PBT

Tm (°C)

65.3

56.6

29.3 / 93.3

149.2

191.4

222.4

ΔHf (J/g)

43.6

35.7

1.9 / 6.3

11.2

20.5

40.7

Tg (°C)

- 29.8

- 34.7

- 38.8

- 32.2

- 21.3

65.3

Tc (°C)

41.0

30.5

22.0 / 1.5

79.0

168.6

180.7

ΔHc (J/g)

68.4

58.0

3.4 / 2.8

17.4

29.6

47.7

(R2-12) In figure 6 labeling of y-axes is missing again.

(Answer) We added y-axis, Heat flow(w/g).

(R2-13) The figure in the paper lack a consistent style regarding font style and font size. For most figures font sizes could be larger, in comparison font size in figure 7 seems gigantic. 

(Answer) Thank you for your format corrections. We modified the font size through all Figures including Figure 7.

Reviewer 3 Report

The aim of presented studies was the synthesis and basic characteristics of poly(butylene sebacate-co-terephthalate) copolyesters with different sebacic acid to dimethyl terephthalate ratio. The motivation was driven by previously reported research on such copolymers confirming their biocompatibility and ability to degradation, performed by other group (Polym Deg and Stab 135 (2017) 18-30). The authors applied the other approach to the melt polycondensation by: first esterification of sebacic acid with 1,4-butanediol, then dimethyl terephthalate with 1,4-butanediol at one stage (if I understand well), second polycondensation under reduced pressure. Unfortunately the synthesis description is very poor to overview the synthesis details. The obtained materials were characterized as regards their chemical structure, molecular weight, thermal and mechanical properties as well as crystalline structure.  The presented research contribute to the bio-based polymers technology, and have a potential to be published, however, in my opinion, Authors have not put enough effort to ensure high quality of the paper. It’s rather a report than scientific paper. There is not deeper discussion about the nature of the copolymers or structure – properties relationships, what is expected from a scientific study. I think the paper should be improved and the discussion could be include in Conclusions section. Below please find my detailed comments to different sections:

  • Introduction is too long, paragraphs 2 and 3 (lines 38 and 45) should be combined in one. Authors should also emphasize what is a novelty in their studies if compared to previously published papers (e.g. Polym Deg and Stab 135 (2017) 18-30);

lines 73-74: butanediol is not a dicarboxylic acid.

  • Synthesis (Section 2.2): the description is too poor, there is no info about the synthesis equipment used, amounts of reagents, catalysts, process parameters and duration. All these data should be included.
  • Results and discussion (Section 3)

Table 2: was the melt flow index measured only once for every material? What is the accuracy of the GPC technique? The numbers are too accurate. 

Mechanical properties: could you explain how the samples elongation was measured? Did you use an extensometer to evaluate the samples extension only in measuring area? The elongation values are very high, what is always worth to validate. In order to discuss about rubber-like characteristics of materials the stress – strain curves should be provided, large elongation is not the only determinant of rubber-like elasticity.

Thermal properties: DSC analysis of polymer materials (particularly new materials) should be performed on the equipment which is able to detect the most characteristic transition temperatures (Tg, Tm, Tc). Please repeat DSC tests in the whole temperature range to fully analyse the exo- and endothermic effects and determine their enthalpies. And then please discuss the thermal properties of investigated materials.  

Crystalline properties: WAXS analysis is very sensitive for a presence of crystalline structures in polymers, even if they seem to be completely amorphous according to DSC. In your materials PBSeT46 and 64 also the crystalline peaks can be distinguished, so the samples are not fully amorphous. In this section Authors mentioned that WAXS enables “degree of crystallization” determination, but no data concerning crystallinity degree of investigated materials are provided. Moreover the title of Section 3.6 is not proper, it rather should be “The crystalline structure”.

Please make a careful revision of the manuscript due to some mistyping.       

Author Response

Thank you for your kind and valuable comments for this study. Reviewer’s comments were very useful and helpful to understand more in detail information and discussion. We tried to follow all your suggestions and appreciated your comments.

(R3-1)  Introduction is too long, paragraphs 2 and 3 (lines 38 and 45) should be combined in one. Authors should also emphasize what is a novelty in their studies if compared to previously published papers (e.g. Polym Deg and Stab 135 (2017) 18-30);

(Answer) Paragraphs 2 and 3 were combined and shortened. We added sentences to emphasize the differences compared to previous research papers in Lines 72-78. We respect their researches and efforts, and we had many insights from their researches also.

Briefly, we wanted to use DMT instead of PTA, and to increase Mw of copolymers comparable to commercialized bioplastic products. Moreover, we tried to perform mechanical property tests to give some insights to industries as well as academia.

Lines 39-47 : “Therefore, the importance of developing eco-friendly biodegradable plastic materials to solve the problems of conventional plastic materials mentioned above is rapidly increasing. Two major strategies have been used for the development of eco-friendly biodegradable polymer materials. The first involves avoiding the consumption of non-renewable resources by using a polymer monomer obtained from nature [4–6], and the second consists of designing polymers with a biodegradable main chain [7]. Based on these strategies, poly(lactic acid) (PLA), poly(butylene succinate) (PBS), and poly(butylene adipate-co-terephthalate) (PBAT) have been used as biodegradable polymeric materials [8–11], and some non-degradable plastics have recently been gradually replaced by biodegradable plastics [12,13].”.

Lines 72-78 : “Poly(butylene sebacate-co-terephthalate) (PBSeT) with about 19,000 g/mol of molecular weight has been synthesized using different compositions of Se and terephthalic acid, and the most amorphous PBSeT (5:5 sebacic acid/ terephthalic acid mole ratio) sample showed highly promising biodegradable characteristics [24]. However, previous researches reported synthesized PBSeT with relatively low molecular weights and only few mechanical property tests were conducted. And dimethyl terephthalate (DMT) has to be adequately screened as an aromatic unit, that mostly purified terephthalic acid (PTA) has been used.

(R3-2) lines 73-74: butanediol is not a dicarboxylic acid.

(Answer) Thank you for your detailed comment. We edited up to your comment.

Lines 68-69: “The biodegradable aliphatic polymer poly(butylene sebacate) (PBSe) can also be obtained through the polymerization of Se as a dicarboxylic acid with butanediol”

(R3-3) Synthesis (Section 2.2): the description is too poor, there is no info about the synthesis equipment used, amounts of reagents, catalysts, process parameters and duration. All these data should be included.

(Answer) We edited the synthesis process more in detail such as the amount of catalyst used and vacuum level, and also added the scheme of synthetic route in Figure 1. We had hard time for trying different synthetic routes with different controllable parameters. We could not give some parameters in too detail, but I believe that scientists in this area are able to understand this method. We hope to publish this random copolymers in series.

Lines 91-100: “The synthesis was carried out using two different melting stages with different levels of vacuum (Figure 1). It was conducted by dosing calculated amounts of materials based on 1 mole of dicarboxylic acid. The first melting stage was the esterification stage, the second was the polycondensation stage. The esterification progressed at approximately 200–210 °C. The theoretical amount of by-products was obtained through esterification against dimethyl terephthalate and 1,4-butanediol with the 0.75 (g/mole) TBT catalyst, and sebacic acid was added to the second esterification reaction against oligomer (T-B). The esterification was finished after obtaining the theoretical amount of calculated by-products. After esterification, we initiated polycondensation by increasing the temperature to 240 °C and decreasing the vacuum level (1.5 torr). All experiments were conducted using different molar ratios of sebacic acid and dimethyl terephthalate; the ratio of 1,4-butanediol to dicarboxylic acid was fixed as 1.25 : 1 mol/mol.”

(R3-4) Table 3: was the melt flow index measured only once for every material? What is the accuracy of the GPC technique? The numbers are too accurate. 

(Answer) Thank you for your valuable comment. We tested each sample more than three times to ensure the accuracy and repeatability. We rounded up the averages.

Table 3. Melting flow index, polymer molecular weight and polydispersity index (Mw/Mn) of copolyesters.

Unit

PBSe

PBSeT82

PBSeT64

PBSeT46

PBSeT28

MFI

(Melting flow index)

g(190 °C / 2.16 Kg)

14

19

18.2

14

0.2

Mw

g/mol

175,500

15,300

154,900

121,600

88,700

Mn

g/mol

53,900

54,500

64,600

35,200

26,700

PDI
(polydispersity index)

Mw/Mn

3.2

2.7

2.3

3.4

3.3

(R3-5) Mechanical properties: could you explain how the samples elongation was measured? Did you use an extensometer to evaluate the samples extension only in measuring area? The elongation values are very high, what is always worth to validate. In order to discuss about rubber-like characteristics of materials the stress – strain curves should be provided, large elongation is not the only determinant of rubber-like elasticity.

(Answer) At first, we measured the elongation at break using extensometer equipped UTM, but some samples were not broken even at the full extension of UTM. Therefore, we changed the specimen with the smallest gauge distance, following ISO 527-3 (number 5 shape) with the grip gap as 33mm. The description on ISO 527 10.2.2 Nominal strain part was helpful to design this SS tests without the extensometer, which was not fit into 25mm gap. Finally, we could get the break point. So we mentioned as “The length of the sample between the grips of the testing machine was 33 mm, because we chose the shortest gap owing to the extremely long elongation of the samples.” in Line 146.
We also added Stress-Strain curves (Figure 5), in which PBSeT64 showed good elastomeric characteristic.

(R3-6) Thermal properties: DSC analysis of polymer materials (particularly new materials) should be performed on the equipment which is able to detect the most characteristic transition temperatures (Tg, Tm, Tc). Please repeat DSC tests in the whole temperature range to fully analyse the exo- and endothermic effects and determine their enthalpies. And then please discuss the thermal properties of investigated materials.  

(Answer) We repeated DSC tests following your suggestion and added Tg and enthalpies in Table 5.

Table 5. DSC data of the synthesized polyesters with various sebacic acid : dimethyl terephthalate ratios.

PBSe

PBSeT82

PBSeT64

PBSeT46

PBSeT28

PBT

Tm (°C)

65.3

56.6

29.3 / 93.3

149.2

191.4

222.4

ΔHf (J/g)

43.6

35.7

1.9 / 6.3

11.2

20.5

40.7

Tg (°C)

- 29.8

- 34.7

- 38.8

- 32.2

- 21.3

65.3

Tc (°C)

41.0

30.5

22.0 / 1.5

79.0

168.6

180.7

ΔHc (J/g)

68.4

58.0

3.4 / 2.8

17.4

29.6

47.7

(R3-7) Crystalline properties: WAXS analysis is very sensitive for a presence of crystalline structures in polymers, even if they seem to be completely amorphous according to DSC. In your materials PBSeT46 and 64 also the crystalline peaks can be distinguished, so the samples are not fully amorphous. In this section Authors mentioned that WAXS enables “degree of crystallization” determination, but no data concerning crystallinity degree of investigated materials are provided. Moreover the title of Section 3.6 is not proper, it rather should be “The crystalline structure”.

(Answer) We agreed with you suggestions. We calculated the degree of crystalinity (Figure 8) and add discusion in Lines 363-366. Also title was changed to “3.6 Crystalline structure”. We appriciate your sugestions again. 

Lines 363-366: “Figure 8 also showed degree of crystallinity of synthesized samples. PBSe showed the highest degree of crystallinity as 64.3%, while that of PBSeT 64 was only 6.7%. It revealed that 6 : 4 molar ratio (Se : DMT) for synthesis of random copolymer was suitable for making polymers with mostly amorphous state domain.”

Figure 8. WAXD curves and crystallinity of the synthesized random copolymers with various sebacate : dimetylterephthalate molar ratios. -See attached file

Round 2

Reviewer 3 Report

I'm satisfied with the current version of the paper. Thank you and good luck with further experiments.